Syria; conflict; suicide rates; suicide risk factors; content analysis

**Corresponding author:**
Salma Almidani;
Email: s.a.almidani@gmail.com

# Demographics and risk factors for suicide in Syria: A retrospective media content analysis of online news sources

Salma Almidani[1,2] 🔘, Mohammad Abo Hilal[3], Marwa Alghadban[3],
Omama Abou Helal[3], Manal Alkourdi[3], Juman Kannan[3], Abdulkarim Ekzayez[1,4],
Preeti Patel[1] and Nassim El Achi[1]

[1]Research for Health Systems Strengthening in Syria (R4HSSS) and the Centre for Conflict and Health Research (CCHR), Department of War Studies, King's College London, London, UK; [2]Department of Primary Care, School of Medicine, University of Nottingham, Nottingham, UK; [3]Syria Bright Future, Gaziantep, Turkey and [4]Syria Public Health Network, London, UK

## Abstract

Since the beginning of the Syrian conflict in 2011, Syrians have faced violence and displacement causing an increase in mental health issues. The COVID-19 pandemic, the 2023 earthquake, and deteriorating living conditions have exacerbated these issues. Suicide in Syria remains an under-researched topic since accurate data are difficult to obtain. In this study, we aimed to explore the demographics and risk factors of suicide in Syria by performing a retrospective content analysis of selected online news (media) outlets from across Syria. Twelve news outlets from the three regions of Syria were selected and news of suicide cases were searched retrospectively. The age range was between 9 and 79 years old with the average age being 27.1 ± SD 5.9 years. The most reported causes of suicide were harsh living conditions (18.5%) and relationship problems (18.3%). The most common method of suicide was hanging followed by using firearms. More suicides occurred at night and in the summer and spring seasons. Based on our study's results, young adult, male, unmarried, individuals in rural settings and northern governorates were at the highest risk of suicide in Syria. This study highlights the urgent need for mental health interventions that address the unique challenges faced by Syrians.

## Impact statement

This study addresses a significant gap in the literature on suicide in Syria, a country that is still suffering from the consequences of one of the most dreadful conflicts of the century. Due to the lack of a suicide surveillance system in the country, along with the stigma and shame associated with suicide in the Syrian context, suicide is highly underreported and under-researched. Therefore, we aimed in this study to explore the demographics and risk factors of suicide in Syria by performing a retrospective content analysis of selected online news (media) outlets from across Syria. We are using the same approach used by scholars in other settings that also lack reliable suicide surveillance systems or reporting such as Bangladesh and Nigeria. The manuscript will be of interest to readers of the journal, as it presents evidence on a critical consequence of mental health issues at a time when researchers, universities, funders, and policymakers are increasingly interested in strengthening and investing in health systems and health research in areas affected by conflict. Such studies can pave the way for integrating mental health as part of health policies intended for health systems strengthening in such contexts.



## Introduction

Globally, more than 700,000 deaths by suicide are reported each year with 77% of those deaths in low- and middle-income countries (LMICs; World Health Organization Factsheets, 2023). According to the World Health Organization (WHO) in 2019, Syria's average suicide rate was 1.95 deaths per 100,000 people (World Health Organization, 2022). This is compared to the global average of 9 suicides per 100,000 and 5.85 suicides per 100,000 in the Eastern Mediterranean region (World Health Organization, 2022). However, underreporting of suicides in countries with religious populations, like Syria, is common and the true number could be much higher (Pritchard et al., 2020). Recent reports highlight a concerning trend of increasing deaths by suicide in northwest Syria with a total of 246 suicides and 1,748 attempts recorded in the final months 2020, surging from 132 recorded suicides in the first quarter of the year (Save the

Children, 2021). One in five documented suicide attempts and deaths by suicide were children (Save the Children, 2021).

Suicidal ideation in northwest Syria is on the increase and reports attribute it to the deteriorating living conditions in the region (Jourdi and Kyrillos, 2022). Social stressors such as poverty, displacement, malnutrition, and the breakdown of social networks are exacerbated by conflict and negatively impact the mental health of individuals (Miller and Rasmussen, 2010). These reports also indicate that young people, especially those under 18, have been particularly prone to suicide ideation (Jourdi and Kyrillos, 2022). These reports align with some of the earliest writings on suicide and conflict by sociologist Emile Durkheim, who proposed that during civil wars there is an increase in suicides due to social disintegration, feelings of hopelessness, and despair among the population (Durkheim and Simpson, 2002). Syria is located on the east coast of the Mediterranean in the Middle East with a population of approximately 23 million people (United Nations, 2022). Since the beginning of the Syrian conflict in 2011, the conflict caused more than 875,000 deaths (Alrashid Alhiraki et al., 2022), approximately 8 million Syrians became refugees in other countries (Abbara et al., 2022), and approximately 6,754,277 Syrians have been internally displaced (UNHCR, 2022). The country has spiraled into a humanitarian catastrophe with violence and displacement causing an increase in trauma and mental health issues (Kakaje et al., 2021). Additionally, the country remains divided between (1) Syrian regime-controlled areas in the majority of the country including major cities and the capital Damascus, (2) opposition-controlled areas in the northwest, and (3) Syrian Democratic Forces (SDF) controlled areas in the northeast (European Union Agency for Asylum, 2020). The coronavirus disease-2019 (COVID-19) pandemic and deteriorating economic and living conditions have exacerbated these issues. According to numbers reported to the WHO, the COVID-19 pandemic resulted in 3,163 deaths in Syria however, studies argue that this number is an under-ascertainment due to limited testing capacity and other factors (Watson et al., 2021; Syrian Arab Republic: WHO, 2023). Mental healthcare before the war was limited, but the documented attacks on healthcare facilities by the Syrian government and its allies have greatly affected access and delivery of mental healthcare (Ri et al., 2019; Hamza and Hicks, 2021). In Syrian society, suicide and suicide attempts remain intertwined with stigma, shame, and social exclusion (Kakaje et al., 2021). Additionally, the criminalization of attempting suicide in Syrian Law plays a role in the lack of suicide surveillance mechanisms in the country (Hassan and Kirmayer, 2015). Suicide in Syria remains an under-researched topic since accurate data are difficult to obtain. Therefore, in this study, we aimed to explore the demographics and risk factors of suicide in Syria by performing a retrospective content analysis of selected online news (media) outlets from across Syria. By identifying the demographic and risk factors for suicide, future research can build on this study to help better inform local and international stakeholders' efforts to address suicide and related mental health risk factors in Syria. Additionally, understanding suicide risk factors can help researchers and policymakers identify commonalities or unique factors that may contribute to suicidal behaviors in the Syrian context, which could be applicable in other regions facing similar challenges.

## Methods

### Data collection

This study followed the methodology of Arafat et al. (2018) in Bangladesh where the situation is similar to Syria and a suicide surveillance system does not exist. Suicide surveillance to obtain accurate data remains a challenge globally, but LMICs specifically often lack the capacity to initiate and sustain surveillance systems (Shah et al., 2017; Arafat et al., 2018). Multiple studies in Bangladesh have relied on news sources and content analysis to understand suicide risk factors due to the lack of suicide surveillance or any nationwide suicide study (Shah et al., 2017; Arafat et al., 2018). This methodology has also been used to study suicidal risk factors in specific populations in Bangladesh like medical students (Mamun et al., 2020), and in countries where suicide surveillance is also lacking like Nigeria (Oyetunji et al., 2021).

### Selection of news sources

Despite the editorial biases, we opted to review online media only due to the feasibility of accessing and searching archived material from previous years for the content analysis. Furthermore, these online sources are widely used by Syrians throughout Syria, which allows us to collect more diverse and inclusive demographic information across regions and areas of control. Before selecting the main online news outlets for the study, three researchers from the team simultaneously screened 34 outlets from the three regions of Syria (regime-controlled, opposition-controlled, and northwest) using search engines. During the screening, the authors used search terms "suicide," "he committed suicide," and "she committed suicide" per the terminology commonly used in the Arabic language to describe death by suicide. From the screening, they selected 12 news outlets that published suicide news with a frequency of more than 30%. Included news sources are Aleppo Today, Sy24, Rozna, and Baladi (opposition-controlled) SnackSyrian, Alkhabar, and Al-watan Online (government-controlled) Vedeng, North press agency, Rumaf, Euphratespost, and Radio Alkul (north-west). These news outlets include a mix of digital media such as online TV channels, news platforms, and magazines and are widely known in their respective regions. The news sources vary in their independence, where news outlets in regime-controlled areas, like Al-Watan, are owned and financially backed by individuals close to the regime therefore they generally promote pro-government content (Trombetta and Pinto, 2023). Many outlets in opposition-controlled areas and in the northwest were only established after 2011 and vary in content and objectivity. The data collected from the news sources were extracted into Microsoft Excel 2019 software for analysis. All articles included were published in Arabic and the extracted data were translated into English. Articles not published in the 12 final sources but published in the 22 sources screened with a suicide news frequency of less than 30% were manually added. Titles were extracted and articles were organized chronologically from the most recent. Duplicates were removed after data were extracted from multiple articles covering the same incident. Articles were included between 1 January 2016 and 31 December 2022.

### Variables

Age, sex, marital status, educational status, occupation, governorate, location (rural, city), and displacement status (local, displaced, refugee camp) were considered as demographic variables in this study. We chose to focus on these variables because they were often included in media reports, but also to examine any associations between suicide and age-related patterns, gender differences, relationship status, educational attainment, certain occupations, regional variations, urban–rural disparities, and displacement.

**Table 1.** Distribution of demographic variables (*n* = 334)

| Demographic variable | Frequency | Percentage |
|---|---|---|
| Age in years | | |
| 0–9 | 1 | .30 |
| 10–19 | 97 | 29.0 |
| 20–29 | 76 | 22.8 |
| 30–39 | 36 | 10.8 |
| 40–49 | 22 | 6.6 |
| 50–60 | 16 | 4.8 |
| Over 60 | 8 | 2.4 |
| Missing | 78 | 23.3 |
| Sex | | |
| Male | 232 | 69.5 |
| Female | 101 | 30.2 |
| Missing | 1 | 0.3 |
| Marital status | | |
| Single | 67 | 20.0 |
| Married | 59 | 17.7 |
| Divorced | 4 | 1.2 |
| Widowed | 2 | 0.6 |
| Missing | 202 | 60.5 |
| Educational status | | |
| Primary | 1 | 0.3 |
| Secondary | 12 | 3.6 |
| High school | 9 | 2.7 |
| University | 12 | 3.6 |
| Postgraduate | 1 | 0.3 |
| Missing | 299 | 89.5 |
| Location | | |
| Urban | 76 | 22.8 |
| Rural | 155 | 46.4 |
| Missing | 103 | 30.8 |
| Occupation | | |
| Laborer | 14 | 4.2 |
| Soldier | 8 | 2.4 |
| Teacher | 4 | 1.2 |
| Police officer | 4 | 1.2 |
| Office employee | 2 | 0.6 |
| Artist | 3 | 0.9 |
| Priest | 1 | 0.3 |
| Drug dealer | 1 | 0.3 |
| Partner in a pharmaceutical company | 1 | 0.3 |
| Shepard | 1 | 0.3 |
| Currency trader | 1 | 0.3 |
| Healthcare worker | 2 | 0.6 |

(*Continued*)

**Table 1.** (Continued)

| Demographic variable | Frequency | Percentage |
|---|---|---|
| Not mentioned | 292 | 87.4 |
| Displacement status | | |
| Local | 171 | 51.2 |
| Displaced | 24 | 7.2 |
| Refugee camp | 13 | 3.9 |
| Other | 6 | 1.8 |
| Missing | 120 | 35.9 |
| Governorate | | |
| Aleppo | 67 | 20.1 |
| Hasaka | 44 | 13.2 |
| Idlib | 41 | 12.3 |
| Homs | 25 | 7.5 |
| Damascus | 23 | 6.9 |
| Hama | 21 | 6.3 |
| Lattakia | 23 | 6.9 |
| As Suwayda | 27 | 8.1 |
| Tartous | 20 | 6.0 |
| Deir ez–Zur | 13 | 3.8 |
| Rif–Dimashq | 6 | 1.7 |
| Raqqa | 5 | 1.5 |
| Daraa | 2 | .6 |
| Al Qunaitra | 1 | .3 |
| Missing | 16 | 4.8 |

Year, season, month, time of attempt, method of suicide, location of suicide, reason for suicide, and presence of suicide note were considered as suicide variables. These variables could help identify avenues for interventions on access to suicide methods, temporal and seasonal patterns, and socioeconomic factors.

## Results

A total of 334 cases were included in the study from the 12 selected online news outlets from 1 January 2016 to 31 December 2022. The age range was between 9 and 79 years old with the average age being 27.1 years ±SD 5.9 years. Since the cases were extracted from online news sources, many of the demographic and suicide variables are missing from the reports, which are reflected in Tables 1 and 2. Suicide was most reported among ages 10–19 at 29.0% followed by 20–29 at 22.8% and ages 30–39 at 10.8% (Table 1). Most were male at 69.5% while women made up only 30.2% of cases (Table 1). Most cases were unmarried individuals 20.0% compared to 17.7% who were reported as married (Table 1). For cases where educational status was reported, most had a university education (3.6%) or secondary education (3.6%) with a large proportion of this variable missing. In terms of occupation, 4.2% of cases were laborers, 2.4% were soldiers, 1.2% were teachers and the same percentage were police officers (Table 1). More cases were reported in the rural setting (46.4%) than in the urban setting (22.8%) (Table 1). Displaced individuals made up only 7.2% of cases with reported resident status in addition to 3.9% of refugees living in camps while most reported cases were local residents 51.2%. The governorates with the largest percentage of reported suicide cases were Aleppo 20.1%, Hasaka 13.2%, Idlib12.3%, and As Suwayda 8.1% (Figure 1).

The most reported cause of suicide was harsh living conditions (18.5%) followed by relationship problems (18.3%). The cases classified as caused by "harsh living conditions" either had stated the Arabic phrase "harsh living conditions" or mentioned poverty, poor financial situation, inability to provide food for family, and other similar descriptors. The cause was classified as "relationship problems" by the research team if the article mentioned familial, spousal, or relationship issues such as "young man and young woman die by suicide in the town of Jarda, east of Deir ez-Zur governate after parents refuse to let them marry and force the young woman to marry an older man." Approximately 33.8% of the suicide deaths were by hanging, 24.2% by use of a firearm, and 15.0% by falling from heights (Table 2). The most common place of death by suicide was in the home at 38.0% followed by a public facility at 11.7%. Only 5.4% of the cases mentioned the presence of a suicide note with the majority of suicide cases (94.6%) having no mention of a suicide note (Table 2). More suicides occurred at nighttime (12.3%) than in the daytime. Most of the suicide cases reported occurred in the summer at 28.7% and a similar percentage of cases occurred in spring and winter at 21.9 and 21.6%, respectively. Based on the collected reports, 2022 had the most suicides at 30.2% then 2021 at 23.1% and 2020 at 21.9% (Figure 2). More

**Table 2.** Distribution of suicide variables (*n* = 334)

| Suicide variable | Frequency | Percentage |
|---|---|---|
| Reason for suicide | | |
| Hard living conditions | 62 | 18.5 |
| Relationship issues | 61 | 18.3 |
| Psychological issues | 28 | 8.4 |
| School–related problems | 15 | 4.5 |
| Video game | 8 | 2.4 |
| Not mentioned | 136 | 40.7 |
| Military draft | 5 | 1.5 |
| Threats/blackmail | 4 | 1.2 |
| Physical illness | 4 | 1.2 |
| Loss of a loved one | 3 | 0.9 |
| Sexual assault/rape | 3 | 0.9 |
| Crime | 3 | 0.9 |
| Imitating TV show | 2 | 0.6 |
| Methods | | |
| Hanging | 113 | 33.8 |
| Firearm | 81 | 24.2 |
| Fall from heights | 50 | 15.0 |
| Poisoning | 34 | 10.2 |
| Hand grenade | 16 | 4.8 |
| Self–immolation | 7 | 2.1 |
| Sharp object | 1 | 0.3 |
| Electrocution | 1 | 0.3 |
| Missing | 31 | 9.3 |
| Location of suicide | | |
| Home | 127 | 38.0 |
| Public facility | 39 | 11.7 |
| Building | 12 | 3.6 |
| Agricultural land | 12 | 3.6 |
| Tent | 8 | 2.4 |
| University | 5 | 1.5 |
| Military checkpoint | 3 | 0.9 |
| School | 1 | 0.3 |
| Missing | 127 | 38.0 |
| Suicide note | | |
| Yes | 18 | 5.4 |
| No | 316 | 94.6 |
| Time of attempt | | |
| Night | 41 | 12.3 |
| Day | 33 | 9.9 |
| Missing | 260 | 77.8 |
| Season | | |
| Summer | 96 | 28.7 |

(*Continued*)

**Table 2.** (*Continued*)

| Suicide variable | Frequency | Percentage |
|---|---|---|
| Spring | 73 | 21.9 |
| Winter | 72 | 21.6 |
| Fall | 68 | 20.3 |
| Missing | 25 | 7.5 |

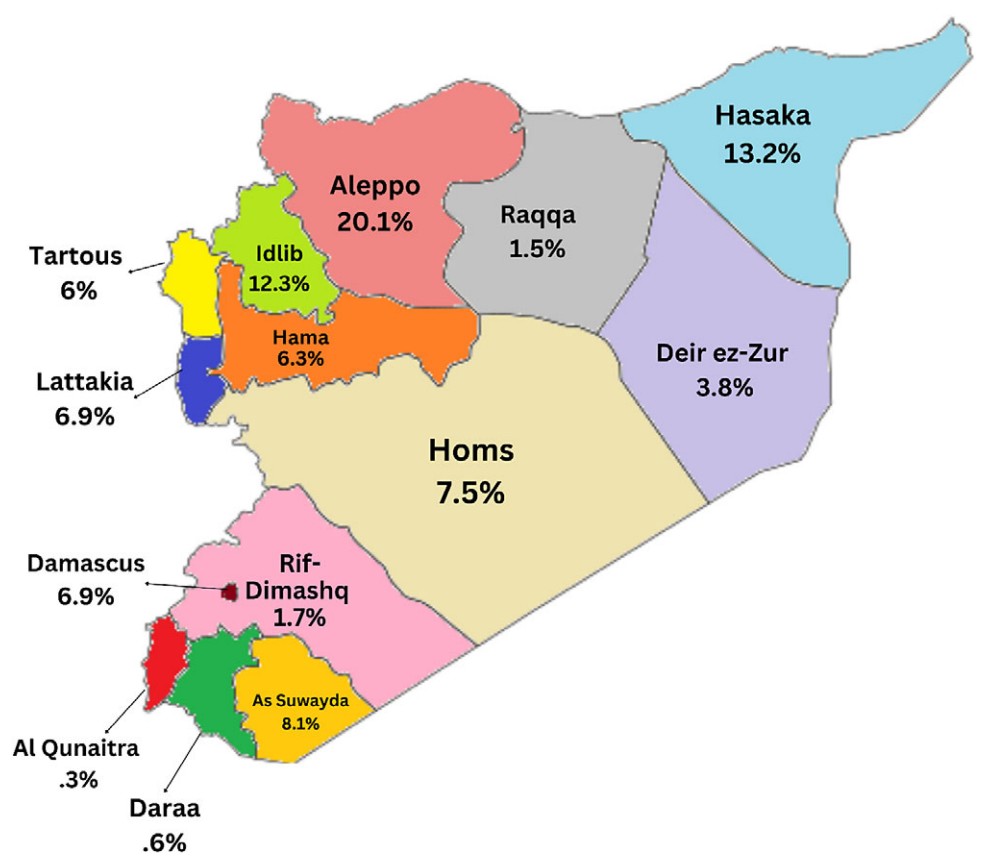

**Figure 1.** Percentage of reported suicide cases in Syria by governorate.

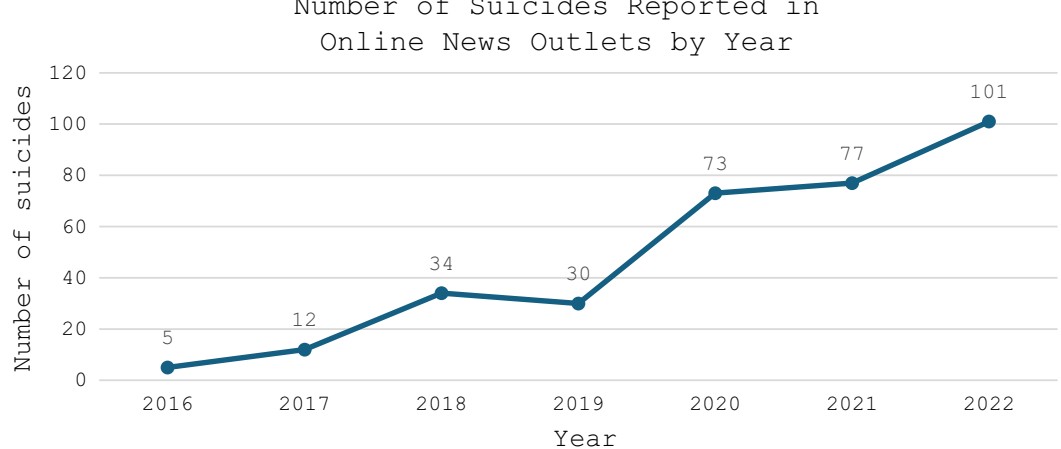

**Figure 2.** Number of suicides reported in Syrian online news outlets by year.

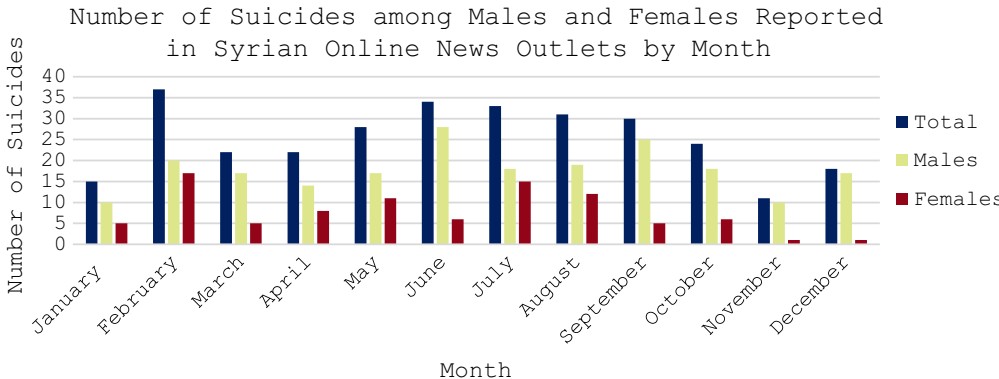

**Figure 3.** Number of suicides reported in Syrian online media outlets by gender and month.

suicides were reported in February (11.1%) than in any other month followed by June (10.2%), July (9.8%), and August (9.3%) (Figure 3).

## Discussion

Suicide in Syria remains an under-researched topic and the aim of this study was to understand the demographic and suicide risk factors by retrospectively analyzing online news portals. The results showed that the average age of suicide cases is 27.1 ± SD 5.9 years old. The results align with WHO data on crude suicide rates where the age group 25–34 years old had the highest suicide rate at 2.95 suicides per 100,000 people (World Health Organization, 2022). However, the age group with the highest number of suicide cases reported was ages 10–19, which reflects the observed increase in suicides and suicidal ideation reported by non-peer-reviewed studies conducted primarily in northwestern Syria (Save the Children, 2021). The data also align with global trends where the leading cause of death among children under 15 years of age and the second leading cause of death among youth 15–29 years old (Biswas et al., 2020). Syrian youth are vulnerable to sources of distress such as trauma, exploitation, early marriage, and stressors of conflict and displacement, which impacts their psychological well-being and could lead to suicidal ideation or suicidal behavior (Colucci et al., 2022). This finding also highlights that special considerations for children and youth-focused interventions should be prioritized in the Syrian context. Previous interventions utilized humanitarian workers to deliver suicide first aid to children and adolescents, however as the living conditions deteriorate in the region more needs to be done to create immediate referral pathways for children/youth at suicide risk and interventions that focus on socio-emotional life skills and delivery for this age group (Colucci et al., 2022).

According to the WHO data in 2019, Syrian males had a higher suicide rate than females, which is similar to the findings of our study where males accounted for 69.5% of cases compared to 30.2% for females. This reveals that men in Syria could be at higher risk of dying by suicide, which could be due to the societal expectations of Syrian men to provide for their families despite hard living conditions in addition to the stigma intertwined with speaking about mental health issues and seeking help. Reports have attributed an increase in suicidal ideation and suicide among Syrians in northwest Syria to the worsening living conditions in the country (Save the Children, 2021; Jourdi and Kyrillos, 2022). These reports align

with the results of our study showing harsh living conditions as the most common cause for suicide.

Research from previous conflict-affected areas such as Croatia showed that high rates of unemployment, lower standards of living, social stressors, and psychological factors all coupled with conflict had a potential impact on the increase in suicide rates (Grubišić-Ilić et al., 2002; Bosnar et al., 2004). Similarly, studies in Kosovo implied a link between the elevated levels of suicidal ideation in the population and the high rates of posttraumatic stress disorder (PTSD) and hopelessness due to the conflict (Wenzel et al., 2009). As living conditions continue to deteriorate and the conflict enters its 13th year, Syrians will continue to experience stressors leading to a potential increase in suicide attempts. This is especially alarming as younger children are exhibiting suicidal thoughts and self-harm behavior, which should prompt investment in suicide research and development of evidence-based interventions like raising awareness, capacity building, and surveillance (World Health Organization, 2021). Unmarried individuals with a secondary or university education appeared to be at higher risk of suicide whereas previous studies have shown that social ties such as marriage can be a protective factor against suicide risk (Øien-Ødegaard et al., 2021). However, educational status is not associated with lower risk of suicide in our sample, but due to the large missing educational data for cases these conclusions must be interpreted cautiously. The study also revealed more suicides occurred in rural areas than in urban areas, which is a trend observed in the literature, but often attributed to other factors related to rural life like isolation, ease of access to means of suicide, and lack of mental health resources (Casant and Helbich, 2022). Studies in post-war Kosovo also found higher rates of suicidal ideation among those living in rural areas compared to urban areas (Wenzel et al., 2009). Researchers have also implied that those with occupations that work outdoors have a higher suicide rate than those who work indoors, which is confirmed by our study data where laborers and soldiers had the highest number of suicides (Casant and Helbich, 2022). In terms of displacement status, the majority of reported suicides were residents of the area followed by displaced individuals, followed by refugees living in camps. Little data on suicidality among refugees and displaced people exist, especially those internally displaced and living in LMICs (Ager et al., 2021). The governorates with the highest number of suicide cases were Aleppo, Hasaka, and Idlib all of which are in northern Syria (Figure 1). Northwest and northeast Syria has experienced a declining economic situation in addition to food shortages and limited access to essential items. These regions host large populations of displaced individuals, most of them

women and children (UNHCR, 2022). The living conditions in addition to the lack of opportunities, the hopelessness, and the stressors of life in conflict can all compound the risk factors for experiencing suicidal behavior (Save the Children, 2021; Jourdi and Kyrillos, 2022).

The most commonly reported suicide method was hanging, which is also one of the leading methods globally and studies have found that it is also the leading method of suicide for the East Mediterranean region (Morovatdar et al., 2013; World Health Organization Factsheets, 2023). Hanging was followed by firearms, which also reflects global and regional trends; however, more cases were reported of falling from heights compared to poisoning. This deviates from studies that state that poisoning (or ingestion of pesticides/toxins) is one of the top three methods of death by suicide (Morovatdar et al., 2013; World Health Organization Factsheets, 2023). However, the increase in firearm use as a method of death by suicide could also be due to the increase in the availability of firearms due to the armed conflict (Grubišić-Ilić et al., 2002; Bosnar et al., 2004; Wenzel et al., 2009). Most of the reported suicide deaths happened in the home followed by public facilities and a minority left a suicide note. More in-depth research into the phenomena of suicide in public facilities can help identify "suicide hotspots" where individuals seek out an accessible public site frequently used for suicide (Cox et al., 2013). Data from the study indicate a peak in suicides during the summer followed by spring and winter, which reflects previous findings of suicides peaking in spring to early summer (Woo et al., 2012). While the specific cause of the seasonal peak of suicides during spring and summer is unknown, some researchers propose a positive association between temperature, sunshine, and suicide (Woo et al., 2012). However, the breakdown by months shows that the month with the largest number of reported suicide cases was February followed by the summer months June, July, and August (Figure 3).

## Limitations

Due to the lack of suicide surveillance systems in Syria, the data collected for this study were only from the selected online news portals and is dependent on the uncertain rigor of news reporting, which is not always up to scientific standards. Missing demographic and suicide risk variables in addition to approximations for some of the variables leads us to approach the results of this study with caution. The study's reliance on online news content introduces the possibility of sampling bias where the online sources selected may not be an accurate reflection of Syria's population especially vulnerable populations like internally displaced persons, ethnic minorities, and other groups who might be less likely to be covered in an online news report. This potentially affects the generalizability of the study's findings to the whole of Syria's population. As previously mentioned, attempting suicide is criminalized by Syrian law, which could also impact the media reporting of suicide cases due to fear of the law, especially in regime-controlled areas. Complex interplay between risk factors, individual characteristics, and broader social determinants of suicide may have not been accounted for in addition to confounding variables not mentioned in the news article. Furthermore, the associations found in this study do not necessarily establish a causal relation between the risk factors identified and suicide.

## Future research direction

The current study primarily relied on retrospective analysis of online news content where future research could incorporate qualitative methods, such as interviews or focus groups with individuals directly affected by suicide, mental health professionals, religious leaders, and community leaders (Kabir et al., 2023). This approach would provide deeper insights into the contextual nuances of suicide risk factors in Syria by capturing personal experiences and cultural perceptions that this study could not address. Future research would also benefit from community-based participatory research approaches, which involve community members in the research process ensuring their voices are heard and that findings lead to interventions tailored to the specific needs of the population (Colucci et al., 2022; Linskens et al., 2023). This is crucial for culturally sensitive and contextually relevant research, especially in fragile contexts like Syria. Previously mentioned reports and the results of our study have emphasized the impact of hard living conditions on suicide, so further studies that explore the relationship between economic instability, poverty, unemployment, and suicide risk in Syria are encouraged (Save the Children, 2021; Jourdi and Kyrillos, 2022). Moreover, studies that examine the availability, accessibility, and quality of mental health services in Syria are needed since assessments of healthcare infrastructure and healthcare professionals can provide a comprehensive understanding of mental health support systems in the country.

## Conclusion

Our study's results showed that young adult, male, unmarried, individuals in rural settings and northern governorates were at the highest risk of suicide in Syria. The most common method was hanging and the most reported reason for suicide was hard living conditions. More suicides occurred at night and in the summer/springtime. These results highlight the urgent need for mental health interventions that address the unique challenges faced by Syrians. We urge the development of a suicide surveillance system and the investment in suicide research to better understand the complex nature of suicide risk in the Syrian population and develop evidence-based interventions to prevent suicide by focusing on social, psychological, and environmental factors. Humanitarian actors are encouraged to engage directly with local communities to gain a better understanding of the risk factors associated with suicide, collaboratively develop solutions, and adopt systematic approaches to address this complex issue. More studies need to be conducted in order to collect large-scale data on suicides in Syria to help understand the risk and protective factors and variables associated with suicidal behavior. Special populations like children or survivors of torture which have unique risk factors that make them vulnerable to developing psychological issues and suicidal behavior need to be studied carefully (Ager et al., 2021).

**Open peer review.** To view the open peer review materials for this article, please visit http://doi.org/10.1017/gmh.2024.47.

**Data availability statement.** Data are already provided as part of the submitted manuscript.

**Author contributions.** S.A., M.A.H., and N.E.A. developed the study conception and took the lead in data analysis and manuscript writing. M.A.G., O.A.H., M.A.K., and J.K. conducted initial searches, reviewed the most relevant sources,

and collected the relevant data for analysis. A.E. and P.P. critically revised the manuscript for important intellectual content. All authors revised several draft versions to reflect feedback from other authors and helped shape the manuscript. All authors read, edited, and approved the final manuscript. S.A. and M.A.H. contributed equally to this work.

**Financial support.** This research is funded by UK aid from the UK government and the National Institute for Health Research (NIHR), Research for Health Systems Strengthening in Syria (131207). The views expressed in this study are those of the authors and do not necessarily reflect the UK government's official policies or those of NIHR.

**Competing interest.** The authors declare that they have no competing interests.

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
