## [Editor Report]

General comments. It is an interesting investigation carried out in Syria. Although the method is interesting, it is important to highlight that not all suicides are reported by newspapers, so we are working with a biased sample and very limited information, all of which is a limitation of the study. It is suggested that the authors modify the title to specify that demography and risk factors are assessed from the media, but not online. Also, it is suggested that you please describe the effect of criminalization and media reporting, and the effect this has on the study. It is necessary to clarify why authors choose only online news outlets.

Specific comments:

- p.5 line 82 please add that this is the reported number (which we know is an under-reporting, including in highly religious countries such as Syria, an information which should also be added)

-p.8 what is the reason and need of these ‘variables’? i suggest to reconsider their inclusion. furthermore, the analytical process is not described.

- 'Displaced individuals made up only 7.2% of cases in addition to 3.9%

187 refugees living in camps while most cases were local residents 51.2%.': what made up the remaining 30%?

-’The most reported cause of suicide was harsh living conditions (18.5%) followed by relationship

191 problems (18.3%)." As we can assume that these labels were given by the authors, it would be useful to describe what they mean by such labels and provide some examples/quotes;

- the finding about children/teenagers being at higher risk should be further discussed and need for child/youth specific suicide prevention interventions (such as the mentioned reference Colucci et al. 2022) emphasized. The authors seem to suggest (e.g. line 242) that mental health interventions are the only way to prevent suicide whereas they should broaden the suggestion to include a variety of interventions, including suicide specific (see Colucci et al. 2022) and others included in LIVE LIFE by WHO.

- line 297 ‘the correlations found in this study’ perhaps associations? or descriptives?

- 311 risk and protective factors;

-the authors should not use the expression ‘committed’ as recommended in suicide research literature;

- Several in-text references are incorrect (e.g. Suicide, Suicide rates etc instead of the authors; sometimes full names are used instead of surnames etc, p. 7 line 182 the author is repeated twice) please check referencing style before resubmission.

---

## [Editor Report]

The authors have made important corrections since the previous version and the manuscript is more robust today. It is certainly a good paper and should be published. Authors are only requested to make some minor changes:

1.- The paper assessed demography and risk factors from news papers while the title suggests to the risk factors only. Please change the title align it throughout the manuscript.

2.- review the following details within the manuscript and correct them:

- p. 12 line 3: Add reference (which I believe is just ‘idem’);

- p.16 l.12 I suggest to add religious leaders;

- consider moving Conclusion at the end;

A final proof-reading is required, e.g. p. 12 l.58 it should be ‘whereas’; p. 14 l22 ‘who works’; and commas before ‘which’.

---

## [Editor Report]

The authors made appropriate changes from the previous version and it is sufficient for it to be published.